A new lattice-based password authenticated key exchange scheme with anonymity and reusable key

Seyhan Kübra 1
Akleylek Sedat akleylek@gmail.com 1 2
1 Department of Computer Engineering and Cyber Security and Information Technologies Research and Development Center, Ondokuz Mayis University Samsun , Samsun , Turkey
2 University of Tartu , Tartu , Estonia
Sadek Rowayda
Electronic publication date: 2024 Jan 31
Publication date: 2024
Volume: 10
Electronic Location ID: e1791
Received 2023 Jul 27; Accepted 2023 Dec 12
Copyright: ©2024 Seyhan and Akleylek
Copyright year: 2024
Copyright holder: Seyhan and Akleylek
License: This is an open access article distributed under the terms of the Creative Commons Attribution License, which permits unrestricted use, distribution, reproduction and adaptation in any medium and for any purpose provided that it is properly attributed. For attribution, the original author(s), title, publication source (PeerJ Computer Science) and either DOI or URL of the article must be cited.
License URL: https://creativecommons.org/licenses/by/4.0/

Keywords: Lattice-based cryptography, Post-quantum cryptography, Password-authenticated key exchange, Bilateral generalization inhomogenous short integer solution, Reusable key, Anonymity, Perfect forward secrecy

Funding: TUBITAK 121R006 This research was supported by TUBITAK under Grant No. 121R006. The funders had no role in study design, data collection and analysis, decision to publish, or preparation of the manuscript.

==============================
In this article, we propose a novel bilateral generalization inhomogenous short integer solution (BiGISIS)-based password-authenticated key exchange (PAKE) scheme for post-quantum era security. The hardness assumption of the constructed PAKE is based on newly proposed hard lattice problem, BiGISIS. The main aim of this article is to provide a solution for the post-quantum secure PAKE scheme, which is one of the open problems in the literature. The proposed PAKE is the first BiGISIS-based PAKE that satisfies anonymity and reusable key features. The bilateral-pasteurization (BiP) approach is used to obtain the reusable key, and anonymity is achieved thanks to the additional identity components and hash functions. The reusable key structure reduces the time in the key generation, and anonymity prevents illegal user login attempts. The security analysis is done by following the real-or-random (RoR) model assumptions. As a result of security examinations, perfect forward secrecy (PFS) and integrity are satisfied, and the resistance against eavesdropping, manipulation-based attack (MBA), hash function simulation, impersonation, signal leakage attack (SLA), man-in-the-middle (MitM), known-key security (KKS), and offline password dictionary attack (PDA) is captured. According to the comparison analysis, the proposed PAKE is the first SLA-resistant lattice-based PAKE with reusable key and anonymity properties.

Introduction

The wireless communication technologies that provide practical and easy usage have enabled the widespread use of mobile devices. The increasing number of users and the openness of communication have brought these devices against various threats in terms of security. These attacks may disrupt network operations and adversely affect the system resources. The lack of security features such as authentication and anonymity will cause information leakage and unauthorized access (Dabra, Bala & Kumari, 2020). With authenticated key exchange (AKE) schemes that provide anonymity, the user is authorized to access mobile devices anonymously. Hence, he/she can securely log in to the server over an insecure channel by protecting the privacy of his/her identity. An adversary cannot use and follow the user’s identity thanks to the anonymity feature. Finally, a secure authenticated session key is obtained during the communication.

In practice, the authentication of mobile devices is carried out using single and multi-factor techniques. Single-factor approaches often utilize low-entropy passwords that are inexpensive and easy to remember. In multi-factor methods, biometric components and various hardware are considered as additional factors to increase security (Ometov et al., 2018). Using these techniques, PAKE schemes were designed for password authenticated key sharing to be used in encrypted communication between the mobile device and the server (Feng et al., 2018). The first examples of the PAKE schemes for traditional cryptosystems were created as a password-authenticated version of the Diffie-Hellman (DH) key exchange (KE). Therefore, the security of these schemes was based on the discrete logarithm problem (DLP) (Bellovin & Merritt, 1992; Boyko, MacKenzie & Patel, 2000). Although DLP cannot be solved with today’s computing power, it can be solved with an algorithm proposed by Shor in the presence of sufficiently powerful quantum computers (Shor, 1994). This development has revealed the need to construct secure versions of KE/AKE/PAKE schemes, which enable secure communication in traditional computing systems and mobile devices for the post-quantum era.

Although the most prominent effort to identify and examine secure algorithms for post-quantum cryptography (PQC) is the National Institute of Standards and Technology (NIST)’s standardization process (NIST, 2022), institutions and organizations such as European Telecommunications Standards Institute, Internet Engineering Task Force, American National Standards Institute, and International Organization for Standardization are also actively working on to be ready for the post-quantum epoch. In addition, many studies have been carried out on analyzing the security of the constructed proposal using different techniques and current technologies (Fernández-Caramés, 2019; Mattsson, Smeets & Thormarker, 2021; Joshi, Bhole & Vaswani, 2022; Radanliev, Roure & Santos, 2023). Up-to-date studies show that lattice-based schemes are at the forefront of post-quantum secure protocol design due to their strong security assumptions and worst-case hardness properties (Peikert et al., 2016). According to the current literature, many lattice-based schemes have been proposed to capture post-quantum security, considering different properties, assumptions, and usage areas.

Unlike traditional schemes, reconciliation structures are one of the important components to be considered in lattice-based KE/AKE/PAKE design (Basu et al., 2023). It is defined as the problem that the shared key cannot be obtained directly due to error terms of hard lattice problems. One of the main challenges in designing lattice-based PAKE is SLA vulnerability that arises with the usage of sending additional information-based reconciliation structures (Bindel, Stebila & Veitch, 2021). Although PAKE schemes were not examined in the PQC standardization processes yet, many lattice-based PAKE schemes have been constructed in the literature for different application areas and communication models. According to the current literature, the number of SLA-resistant lattice-based PAKE scheme is quite limited and it is known that existing ones will also have vulnerabilities (Ding, Cheng & Qin, 2022). So, an SLA-resistant PAKE solution will be one of the suitable candidate to capture post-quantum security of key sharing requirements.

Literature review

Lattice-based PAKE schemes have been constructed into two sub-classes: Smooth Projective Hash Functions (SPHF)-based and hard-lattice problem-based. With the usage of SPHF, it was aimed to eliminate the usage of random oracles in getting strong security features. It has been widely preferred in the PAKE design since it is a variation of a zero-knowledge proof system. In the SPHF-based PAKE schemes, public key is used to obtain the commitment of the password and check the validity of this commitment (Cramer & Shoup, 2002). In the last six years, (Zhang & Yu, 2017; Jheng et al., 2018; Li & Wang, 2019; Wang et al., 2019; Jiang et al., 2020; Li, Wang & Morais, 2020; Yin et al., 2020; Tang et al., 2021; Zhao, Ma & Qin, 2023) schemes were proposed to the literature by using SPHF assumptions. The hard-lattice problem-based ones have been defined by transforming DH-like PAKE ideas into lattice-based primitives. These PAKEs can be divided into two categories in terms of number of phases: one-phase and four-phase. The first adaptation of the one-phase PAKE scheme was proposed in Ding et al. (2017). Then, Xu et al. (2017), Liu et al., 2019, Ren, Gu & Wang (2023), Seyhan & Akleylek (2023) were also constructed by using lattice assumptions with different properties and usage areas.

In this section, we present the current literature on four-phase lattice-based PAKEs since the proposed BiGISIS.PAKE is proposed by following the four-phase PAKE design. In Dabra, Bala & Kumari (2020), a PAKE version of Feng et al. (2018) was constructed for the post-quantum era security of mobile devices. The idea of direct public key authentication from the zero-knowledge authentication scheme (Ding et al., 2018) was followed to obtain SLA resistance. Based on the hardness of the ring LWE (RLWE) problem, the scheme consists of setup, registration, login and authentication, and password update phases. In the proposed four-phase two-party PAKE, Ding functions (Ding, Xie & Lin, 2012) were chosen as a solution to the reconciliation problem. The security analysis of the proposed PAKE, which provides key reusability, anonymity, and PFS, was performed in the RoR model by examining attack resistance such as PDA, SLA, and MBA. In Islam & Basu (2021), a three-party and four-phase lattice-based PAKE scheme for mobile devices was proposed. Based on the hardness of the RLWE problem, the authentication was obtained using a timestamp and password. The reconciliation problem was solved by adding Ding functions (Zhang et al., 2015). The security analysis was performed in the ROM by making attack specific examinations. Due to the key reuse property, in Ding, Cheng & Qin (2022), Dabra, Bala & Kumari (2020) had been proven vulnerable to SLA. An improved SLA-resistant RLWE-based PAKE was also constructed by adding the practical randomized KE idea (Gao et al., 2018). The scheme was designed as a four-phase, two-party scheme, and the security analysis was also presented in the RoR. SLA and MBA resistances were discussed by considering mobile environment communication. In Kumar et al. (2023), an SLA-resistant PAKE scheme was constructed for mobile devices. The security of the proposed PAKE was based on the RLWE problem. It was designed as a four-phase PAKE for post-quantum secure two-party communication. The security analysis was performed in the ROM. The resistance against some attacks were examined and it also provided PFS, mutual authentication, and anonymity features. In Wang et al. (2023), an efficient smart-card-based PAKE scheme was proposed to obtain post-quantum secure two-factor authentication with reusable key. The main hardness of the proposed PAKE was based on the RLWE problem and used fuzzy-verifier + honeywords approach (Wang & Wang, 2016) to provide resistance against two-factor-based attacks. The resistance against smard-card-based attacks were analyzed with formal security analysis under the ROM assumptions.

Motivation and contribution

PAKE schemes that meet the authentication requirements for key-sharing techniques by utilizing passwords are used in different areas. Thanks to these schemes, high-entropy shared session keys are obtained by using low-entropy and easy-to-remember passwords. With the PQC concept, the need for quantum secure PAKE schemes for different applications has arisen (Ott, Peikert et al., 2019; Hao, 2021). According to the current literature, two main properties, anonymity, and reusable key, are captured with PAKE schemes. The anonymity that gives user privacy and the reusable key features that efficiently use system resources are essential to propose solutions for these requirements. The number of schemes that provide these features is also quite limited. We sought an answer to how to design a PAKE protocol with appropriate processing steps for post-quantum security of mobile devices with additional features. Then, we constructed a four-phase BiGISIS.PAKE scheme, whose security is based on the assumptions of BiGISIS. The main contributions of the proposed BiGISIS.PAKE to the literature is highlighted as follows.

• The first BiGISIS-based PAKE scheme, which provides anonymity and reusable key features, is proposed for post-quantum security of mobile devices.

• The BiP approach is used to reduce the time spent on key generation by obtaining a reusable key. Unlike the methods in the literature, the reusable key feature does not cause SLA thanks to the part selection-based reconciliation method, the most significant bit (MSB).

• Anonymity is achieved to prevent illegal user login attempts by using pseudo-identity components and additional hash functions.

• The security analysis is presented by following the RoR model assumptions. As a result of security analysis, PFS and integrity are satisfied, and the resistance against eavesdropping, hash function simulation, manipulation, impersonation, MitM, KKS, and offline PDA are captured.

• To the best of our knowledge, it is the first SLA-resistant lattice-based PAKE scheme that provides anonymity and reusable key.

Organization

The rest of this manuscript is organized as follows. In ‘Preliminaries’, the notation and the mathematical background are given. In ‘Proposed PAKE’, each phase of the proposed scheme is explained step-by-step. Parameter selection constraints, basic computations, and protocol flow are explained in detail. In addition, the verification is done by examining basic assumptions and key parameters. In ‘Security Analysis’, the semantic security and attack resistances are discussed. The comparison with similar four-phase lattice-based PAKE schemes is given in ‘Comparison’. Finally, the conclusion is discussed in ‘Conclusion’.

Preliminaries

The symbols used in this article are listed as follows.

• aT: The transpose of the vector a.

• ℜ:ℤ[x]/(xn + 1), ℜq:ℤq[x]/(xn + 1).

• m ≥ 2: Module dimension.

• a ∈ U(x): a is randomly chosen from the distribution x.

• A∈ℜqm×m: m × m-dimensional matrix A is defined in ℜq.

• ś: Server. m´: Mobile user.

• σ=αnl/lognl14: Standard derivation.

• χ = Dℜm,σ: Discrete Gaussian Distribution is defined in ℜm with σ.

• ∣∣: The concatenate operator.

• H{i=0,1,2}(a1, a2, …, a∗): Hash functions produce the outputs according to the specification, detailed in Proposed PAKE section. The input is determined by concatenating sub-components.

• X←rχ: X is randomly chosen from χ.

• pwa, ida: The password and the ID value of a, respectively.

• neg(n): Let ϖ > 0 and n > n0. If there is an n0 ∈ ℕ such that neg(n) < n−ϖ, neg is called negligible function.

The post-quantum security of the proposed PAKE is captured by following the hardness assumption of the BiGISIS problem. The main properties of this problem are recalled with Definition 1.

Definition 1 BiGISIS Problem; Jing et al., 2019

Let x1 = As1 + e1modq and x2T=s2TA+e2Tmodq is computed by using A∈ℜqm×m, s1,s2T←rχ, and e1,e2T←rχ. The main purpose of the BiGISIS problem is to obtain the secret keys s1 and s2T, given the public keys x1 and x2T.

The hardness of the BIGISIS problem is explained by the decisional version of the BiGISIS (DBiGISIS) problem. Let A,x1,x2T be given. The DBiGISIS is a problem of deciding whether x1,x2T belong to the uniformly random distribution (c1=x1,c2T=x2T∈Uℜqm×m×Uℜqm×Uℜqm) or the BiGISIS (x1,x2T∈ BiGISIS) distribution.

The hardness of the DBiGISIS is remembered in Lemma 1.

Lemma 1 Let |PrPPAA,x1,x2T=1−PrPPAA,x1′,x2T′=1|<negn be for any probabilistic polynomial time algorithm (PPA). Then, with a negligible probability, the PPA algorithm can distinguish the distributions K1=A,x1,x2T∈BiGISIS and K2=A,x1′,x2T′∈Uℜqm×m×Uℜqm×Uℜqm (Jing et al., 2019).

The hardness of the DBiGISIS problem was explained by using the M-LWE problem.

Conclusion 1 Assume that the hardness assumption of the DBiGISIS problem given by Lemma 1 is satisfied. Then, the hardness of the DBiGISIS problem is equivalent to the hardness of the decisional-M-LWE problem (Jing et al., 2019).

Remark 1 In Langlois & Stehlé (2015), it was shown that there is no polynomial time algorithm to solve the decisional-M-LWE problem, even if in the presence of quantum computers. Based on these proofs, there is also no PPA to solve the DBiGISIS problem since there is a reduction between decisional-M-LWE.

So, if a scheme is built based on the BiGISIS hardness assumption, it will be secure unless an algorithm can also solve the MLWE problem.

The MSB function was naturally used to agree on the shared key in the BiGISIS-based KE schemes due to the bilateral structure of distribution. In the proposed PAKE, the MSB function is chosen as a reconciliation mechanism to solve the problem of not being able to generate shared keys directly. The explanation of MSB is recalled in Definition 2.

Definition 2 MSB Function

Let x ∈ ℤq be input. There are two possible outputs for y = MSB(x). If q4<|x|<q2 then y = 1, otherwise y = 0.

In the proposed PAKE, cyber attackers cannot extract the actual identity or monitor any user’s action through the compromised information since it provides anonymity property. The definition of anonymity is remembered in Definition 3.

Definition 3 Anonymity

Anonymity means that the identity of one party is kept confidential even from the other party with whom it is communicating. It seems like an anti-authentication feature as one party wants no one to be able to identify it to protect its privacy. Although anonymity and authentication are two opposite features in practice, the user who wants to remain anonymous also wants to be sure of the other party’s identity (Feng et al., 2018). Cyber attackers cannot use or track the user’s identity thanks to the anonymity feature. So, an anonymous PAKE allows legitimate users to log into the server over an unsecured network without compromising their identity. Anonymous authentication has been one of the features evaluated in the PAKE protocol design, as personal security will be protected in online interactions (Goldberg, Stebila & Ustaoglu, 2013). To achieve anonymity in the PAKE scheme design, the user’s identity is stored anonymously with so-called additional components and some operations.

As an additional property, the proposed PAKE satisfies the reusable key to obtain run-time efficiency. The definition of the reusable key is given in Definition 4.

Definition 4 Reusable Key

In KE schemes, it is desired that the public key be reusable to reduce the time in the key generation and storage costs. Different techniques have been proposed to provide the reusable key feature for lattice-based key-sharing schemes (Akleylek & Seyhan, 2020; Seyhan et al., 2021; Zhang et al., 2015; Ding, Branco & Schmitt, 2019; Gao et al., 2018; Ding et al., 2018). The reusable key feature can make the constructed protocols vulnerable to SLA (Bindel, Stebila & Veitch, 2021). An SLA can occur with the reuse of the keys in protocols that use an additional information-based reconciliation mechanism since the shared key and additional information can leak information about the secret key (Qin et al., 2022).

In the literature, the reusable key in lattice-based PAKE schemes was captured with (Ding et al., 2018) and (Gao et al., 2018) methods. However, these techniques caused SLA since they were used with sending additional information-based reconciliation. The BiP method is included in the proposed PAKE to provide the reusable key feature.

Definition 5 BiP Method

Let A∈ℜqm×m, g2,g1T←rχ, H2:{0, 1}∗ → χ be a hash function, and x1,x2T is generated by consideringDefinition 1. The BiP components is defined with x1↔=x1+AH1x1+g2 and x2↔T=x2T+H1x2TA+g1T and provide the reusable key feature due to the same distribution properties. There are two possible situations. 1.Ifx1,x2T∈BiGISIS,thenx1↔,x2↔T∈BiGISIS,2.Otherwise,x1↔,x2↔T⁄∈BiGISIS

The PPA that can solve the BiGISIS problem in case 1 with the BiP method is unknown. In case 2, the components are close to the uniform distribution. The reusable keys are satisfied since any attacker (Ă) cannot obtain information about previous secret keys using these components.

Remark 2 Note that in order to provide the reusable key feature with the BiP method, a reconciliation structure that does not include sending additional information should be used. The SLA-resistant reusable key feature is achieved since the MSB function, which provides reconciliation with the most significant region, is used in the proposed protocol. We refer to Akleylek & Seyhan (2020), Seyhan et al. (2021) to check the detailed proof that shows the reusable key power of the BiP method.

In the proposed PAKE, the CDF-Zipf (Wang et al., 2017) model is followed to characterize the password distribution in the security analysis.

Definition 6 CDF-Zipf Model

Let Correctpw be the event of adversary’s guessing a correct password with the PDA, DS be the size of password dictionary, and nop be the maximum number of active password-guessing attempts before a corrupt query. The probability of event correctpw in the conventional approaches is PrCorrectpw=nopDS+negln. Since these methods underestimate the adversary’s power in real-world applications, CDF-Zipf, which provides the characterized password distribution, is preferred to obtain a realistic examination of password guess. Let C′ ∈ [0.001, 0.1] and f ∈ [0.15, 0.30] be CDF constants that can be computed by linear regression. According to CDF-Zipf, the probability of Correctpw is determined by using Eq. (1). (1) PrCorrectpw=C′⋅nopf+negln

Proposed PAKE

The proposed post-quantum secure two-party BiGISIS.PAKE scheme consists of four phases that ensure anonymity and reusable key features. The main hardness relies on the BiGISIS problem. During the setup phase, system parameters are generated. In the registration phase, registration is made for the interaction of the mobile user with the server. Due to the additional components used in login&password-based mutual authentication, reusable key, authentication, and anonymity features are provided. The possible need for the password update requirement is also defined.

Setup phase

ś performs the setup stage and generates the system parameters.

• n is chosen to be the power of 2. The sensitivity of the application is also considered in the selection of n. Then, the module dimension m is chosen such that m ≥ 2.

• The odd prime number q is selected such that qmod2n = 1.

• The common public key A←rℜqm×m is selected and the distribution χ is defined.

• ś generates the static public key pśT=sśTA+eśT, where sśT,eśT←rχ.

• Three different hash functions H0:{0, 1}∗ → {0, 1}l, H1:0,1l→ℜqm×1, and H2:{0, 1}∗ → χ are specified for the BiP, authentication check, and password-based shared key generation.

At the end of this phase, ś publishes the session parameters n,m,q,χ,A,pśT,Hi=0,1,2.

Registration phase

The mobile user registration is performed after the setup is completed. All messages are assumed to be transmitted over the secure channel at this stage. Due to the registration phase, user registration and the shared key agreement are achieved securely. The mobile user registration is completed by performing the enumerated steps in Fig. 1. Then, the login&password-based mutual authentication phase is started.

Figure 1 The BiGISIS.PAKE registration.

In Fig. 1, the mobile user starts the registration by sending the client ID (idm´) to the server. In lines 3–5, the server makes some random selections (t,sśT∗) and computes the registration parameter (dm´∗) to package the ID and secret key. In line 6, the server sends t,dm´∗,pidm´ to the mobile user to generate registration parameter. In lines 7–9, the mobile user computes the registration parameter (dm´) and packaged password (PWm´ = H1(vm´)). Finally, the mobile user and the server store {pidm´, dm´} and pidm´,PWm´,sśT∗ in the databases, respectively.

Login&password-based mutual authentication phase

At this stage, an insecure version of the communication channel is considered. With password-based authentication, the shared key is generated. By using hash function H0 and cidm´ components, the anonymity feature is obtained. To get the reusable key feature, the BiP method, given in Definition 5, is used. The operations of this stage are detailed in Fig. 2.

Figure 2 Login&password-based mutual authentication.

Let’s detail each party’s computations by looking at them step-by-step according to Fig. 2.

• Lines 1–8 for Mobile User: In line 1, the mobile user recalls the parameters that were generated during the registration phase. In lines 2–3, he/she computes the password (PWm´) to use it in the protocol flow. Between lines 4 and 7, he/she generates static (pm´) and temporary (xm´) public keys by following the BiGISIS distribution. Finally, in lines 8–9, the mobile user computes and sends its packaged public key with the password (xm´∗) to the server.

• Lines 10–21 for Server: In line 10–11, the server decrypts the packaged mobile user public key (xm´∗) with the help of the fetched password (PWm´) by using pidm´. In lines 13–14, the server generates its temporary public key (xśT). In lines 15–17, the server determines and computes BiP components to provide the reusable key feature. Based on these calculations, the server generates its key component (kś). Then, he/she uses MSB reconciliation to agree on the same shared key. Finally, the server sends its public key (xśT) and authentication component (αś) to the mobile user.

• Lines 22–34 for Mobile User: In lines 22–24, the mobile user computes BiP-related components to ensure the reusable key feature with the computation of (xśT↔). In lines 25–27, he/she generates the key component of the server (kś′) based on its computations and determines the reconciliation value (ψś′) to check the first step of successful authentication (αś=?H0xm´,xśT,ψś′). In line 28, he/she determines the pseudo-secret key (sm´) to use it in the second authentication check. Then, the mobile user computes its key component (km´) and generates its reconciliation value (ψm´). In line 31, cidm´ computation is occurred to assist the anonymity feature. Finally, lines 32-33, shared secret key (skm´ś=H0idm´,cidm´,xm´,ψm´,αm´,xśT,ψś′,αś) is generated by using third authentication check component (αm´).

• Lines 35–41 for Server: Like the mobile user, in lines 35–39, the server computes server-related parameters such as key component (km´′), reconciliation value (ψm´′), anonymity item (idm´), pseudo-secret key (sm´), and authentication component (αm´′). Finally, the server also generates the shared key (skś m´=H0idm´,cidm´,xm´,ψm´′,αm´,xśT,ψś,αś) based on generated values.

After running the login&password-based mutual authentication phase, the mobile user and the server agree on a password-authenticated shared key. The final step, the password update phase, is optional and should be used when the parties need to update the password.

Password update phase

The password update step is performed when the user wants to update her/his password. It is assumed that all transmitted messages are sent over an secure channel. The password update stage is explained in Fig. 3. The step-by-step definition of Fig. 3 is summarized as follows.

Figure 3 Password update.

• Lines 1–4 for mobile user: By using fetched {pidm´, dm´}, he/she determines the password component (vm´) to derive the main password (PWm´∗) and sends it to server to check the password update requirement.

• Lines 5–8 for server: The server brings back the stored password (PWm´) by using pidm´. In line 6, if a match is achieved, the server sends a warning message and allows the mobile user to initiate the password update process.

• Lines 9–13 for mobile user: In lines 9–12, the password-related components (pwm´new, dm´new, PWm´new) are re-generated and new password information (dm´new) is stored in the database. Finally, the mobile user sends the updated password (PWm´new) to the server.

• Line 14 for server: The updated password is received from the mobile user and is stored in the server database.

The correctness analysis of the proposed PAKE is discussed in ‘Correctness analysis’.

Correctness analysis

In the proposed BiGISIS.PAKE scheme, the necessary conditions to derive the shared key are computed based on the static and ephemeral key components. Due to the number of sub-components, the upper bound is determined by examining the static ones (kś,kś′). By considering Fig. 2, if kś and kś’ are rewritten, Eq. (2) is obtained. (2) kś′=pś+xś↔Tsm´+rm´+y−pśTsm´+hm´=sśTA+eśT+xśT+zTA+gm´Tsm´+rm´+y−sśTA+eśTsm´+hm´=sśTAsm´+sśTArm´+sśTAy+eśTsm´+eśTrm´+eśTy+rśTAsm´+rśTArm´+rśTAy+fśTsm´+fśTrm´+fśTy+zTAsm´+zTArm´+zTAy+gm´Tsm´+gm´Trm´+gm´Ty−sśTAsm´−eśTsm´+hm´kś=sśT+rśT+zTpm´+xm´↔−sśTpm´+hśT=sśT+rśT+zTAsm´+em´+xm´+Ay+gś−sśTAsm´+em´+hśT=sśTAsm´+rśTAsm´+zTAsm´+sśTem´+rśTem´+zTem´+sśTArm´+rśTArm´+zTArm´+sśTfm´+rśTfm´+zTfm´+sśTAy+rśTAy+zTAy+sśTgś+rśTgś+zTgś−sśTAsm´−sśTem´+hśT

In Eq. (2), for i ∈ {ś, m´}, since si,ri,ei,fi,hi,y,zT∈ℜqm×1, ||si,ri,ei,fi,hi,y,zT||<β=nσ should be satisfied (Jing et al., 2019). The norm of the multiplication of any two of these components is ϑ = mnβ2 (Note 2 in Akleylek & Seyhan, 2020). For instance, ||em´Trś||,||zTfś||≈mnβ2=ϑ. So, reconsidering the relationship between kś′ and kś, Eq. (3) is obtained. (3) k′ś−kś=eśTrm´+eśTy+fśTsm´+fśTrm´+fśTy+gm´Tsm´++gm´Trm´+gm´Ty+hm´−rśTem´+zTem´+sśTfm´+rśTfm´+zTfm´+sśTgś+rśTgś+zTgś+hśT⇒|kś−kś|≤28ϑ+β

Since q = O(2λϑβ) and sśTAsm´≈q, the probability of kś′=kś in the proposed PAKE is at most O(n2−λ).

Additional properties of proposed PAKE

With the proposal of BiGISIS.PAKE, it is aimed to obtain a post-quantum secure PAKE scheme with additional features, basically reusable key and anonymity. These properties are defined in Definitions 3–4. In the constructed scheme, anonymity is captured with the usage of an additional identity component and the one-way property of the hash function. The reusable key is provided by adding BiP components that allow storing the secret key in case the public key is reused. Let’s show how these properties are satisfied with the proposed PAKE according to the login&password-based mutual authentication phase, given in Fig. 2.

• Anonymity: According to Definition 3, idm´ value should not be obtained by the attacker (Ă) to ensure anonymity. In the proposed PAKE, with step 37, idm´=cidm´⊕H0xm´,ψm´′,xśT,ψś is determined. Let’s show that Ă cannot obtain all the components included in this calculation through possible attempts. Suppose that Ă captures {pidm´, xm´}, xśT,αś, and {cidm´, αm´} with MitM attack. Ă obtains xśT∗=rśT∗A+fśT∗ using the modified rśT∗,fśT∗←rχ. Also, by using kś=sśT+rśT+zTpm´+xm´↔−sśTpm´+hśT, Ă can compute ψś∗. Although Ă can get sśT with Corrupt-ś(śj) query, defined in Section ‘Security Analysis’, the BiP method component rśT is unknown. The only possible way to get rśT is to solve the BiGISIS problem. It is known that even in the presence of quantum computers, rśT cannot be obtained as a result of Lemma 1 and Conclusion 1. Overall, Ă cannot get idm´ and anonymity is ensured.

• Reusable Key: In the proposed PAKE, the reusable key is provided with the help of BiP component, added in steps 17 and 24. According to Definition 5, the computed BiP component xm´↔=xm´+Ay+gś and xśT↔=xśT+zTA+gm´T have same distribution properties with xm´ and xśT, respectively and hide the general properties of xm´ and xśT. So, even if an Ă gets skś m´=H0idm´,cidm´,xm´,ψm´′,αm´,xśT,ψś,αś=skm´ś, he/she cannot obtain any information about real/static secret keys when the session is run multiple times. So, the reusable key property is satisfied.

The detailed security analysis of the BiGISIS.PAKE is shown in ‘Security Analysis’.

Security Analysis

The main hardness of the proposed scheme is based on the BiGISIS problem. Since an algorithm that can solve this problem in polynomial time is not known in quantum computers, it provides provable security. The detailed security analysis in RoR (Abdalla, Fouque & Pointcheval, 2005) shows that any PPA Ă can gain a negligible advantage over the scheme. In the RoR model, all standard send, execute, corrupt, reveal, and test queries are considered. Let U be the any party of the scheme.

• send (U, i, M): The message M is sent to the i-th instance U. According to the protocol flow, the instance generates the components and gives the outputs to Ă.

• execute (m´, i, ś, j): The protocol is run between i-th instance m´ and j-th instance ś. The output is the executed protocol transcript and is returned to Ă.

• corrupt (U): The password of instance U is returned to Ă.

• reveal (U, i): This query is made by Ă and obtains the session key of U. The shared key of the i-th instance of user U is sent to Ă as an output. This query is constructed with the requirement that a session key captured by Ă should not affect other sessions.

• test (U, i): i-th instance U flips a bit b. If b = 1, the output is the authenticated shared key of i-th instance U. Otherwise, a uniformly random key is selected as an output.

It is also assumed that a completely random or real shared session key is obtained with an infinite number of test queries. Two additional queries in the Dabra, Bala & Kumari (2020) model are examined to prove that PFS is guaranteed in the proposed scheme. These are remembered as follows.

• Corrupt-m´(m´i): The manipulation situation in which the user device is captured by A is examined. Performing Corrupt-m´(m´i) query assumes that Ă has captured all sensitive information stored on m´’s mobile device.

• Corrupt-ś (śj): With this query, it is assumed that the static secret of ś and the secure database of ś are captured by Ă.

The provable security of a PAKE scheme is examined by dealing with semantic security.

Definition 7 Semantic Security (Abdalla, Fouque & Pointcheval, 2005; Dabra, Bala & Kumari, 2020)

The semantic security is examined by Ă’s success in guessing the bit bobtained by the test query. Let Succeed (S) be the event of Ă correctly guessing b. IfEq. (4) is satisfied, the proposed scheme is said to be semantically secure. (4) AdvĂPAKE=2|PrS−12|<negln

The semantic security of the BiGISIS.PAKE scheme is examined considering Theorem 1.

Theorem 1 Let Ă be the PPA adversary attempting to generate the shared key during the login&password-based mutual authentication phase of the BiGISIS.PAKE scheme. The advantage of Ă (Adv) in breaking the scheme’s semantic security by obtaining the shared key between m´- ś is given byEq. (5). (5) AdvĂBiGISIS.PAKE≤2qH022dH0+qH122dH1+qsend+qexecute22dχ+qsenddiddpw+2Advℜqm×1BiGISIS+C′⋅nopf2

Proof 1 The proof of Theorem 1 is defined by examining the game sequence Gi={0,1,…,6}. Let SGi be the event of correctly guessing b in the test queries of Ă in every game Gi. b is generated by flipping a coin before the game sequence. Then, it is stored in Ă for games containing test queries.

G0:The basic attack is modeled with this game. At the start of the game, Ă guesses the randomly generated b. By rewriting Eq. (4), AdvĂG0=2|PrS0−12| is obtained. In this game, PrS0=12 is procured because the random bit is tried to be guessed. So, AdvĂG0=0.

G1:In G1, the case of an eavesdropping attack on the communication channel of m´- ś is modeled. Because of the eavesdropping in G1, Ă obtains the transmitted information in the scheme by making execute query. This value is used in test queries to determine whether the shared session key is real or random. Let’s examine success of Ă in G1.

In Fig. 2, by eavesdropping, xśT,αś,cidm´,xm´,αm´ are obtained. In the BiGISIS.PAKE scheme, the shared session key is computed with skm´ś=H0idm´,cidm´,xm´,ψm´,αm´,xśT,ψś′,αś=H0idm´,cidm´,xm´,ψm´′,αm´,xśT,ψś,αś=skś m´. In addition to the eavesdropping information, Ă needs some values ({idm´, ψm´, ψś}) to determine the shared session key. Since ψm´=MSBkm´=xś↔Trm´ and ψś′=MSBkś′=pś+xśT↔sm´+rm´+y−pśTsm´+hm´, the computation of these values depends on the static/ephemeral secret key components and idm´. These values cannot be obtained as the algorithm that can solve the BiGISIS problem in polynomial time is not known even in quantum computers. Since Ă does not obtain skm´ś = skś m´, Ă’s success in G1 does not change. It is expressed in Eq. (6). (6) PrSG0−PrSG1=0

G2:In G2, an active attack is modeled by using the hash function, send, reveal, and execute queries. It is mainly based on simulating hash functions and examining their collision properties. In this attack, Ă sends modified messages to parties that cannot be detected without xm´∗,xśT and {αm´, αś} collisions. Let’s examine Ă’s success in G2.

• Ă queries oracles H0 and H1 to find {αm´, αś} collision. H2 oracle is not examined in this game since the parameters of H2 are in the flow of the scheme and can be obtained by execute queries. Let qH0 and qH1 be the number of hash function queries, dH0 = 2l and dH1 = qnm be the output space dimension of H0 and H1, respectively. By considering the birthday paradox, the probability of the collisions of hash functions is maximum qH022dH0+qH122dH1.

• Ă also makes send and execute queries to find xm´∗,xśT collision. In the normal flow, these parameters are calculated with rm´,rśT,fm´,fśT selected from χ, depending on the BiGISIS distribution. Let the output space dimension of the χ be dχ, and the number of execute and send queries are qexecute and qsend. By considering the birthday paradox, the probability of the collisions of xm´∗,xśT is maximum qsend+qexecute22dχ.

• Ă can obtain the session key (skś m´=H0idm´,cidm´,xm´,ψm´′,αm´,xśT,ψś,αś=skm´ś) by using reveal query. In this case, Ă uses these components to examine whether the parties can obtain static and temporary secret keys. Let’s assume that Ă impersonates the mobile user and obtains the shared key. Ă must also know the correct password (PWm´) and the correct static/temporary public keys (xm´, pm´) and authentication components (idm´, sm´) to bypass the authentication control. The probability of obtaining PWm´ is determined as 1dpwdid since Ă should capture the correct ID and PW values, where dpw and did be the dimension of the passwords and ID’s dictionaries, respectively. The correct computation probability of public keys is 2Advℜqm×1BiGISIS since the static and temporary ones can be obtained only by solving the DBiGISIS problem. So, the maximum probability of capturing shared key is 2Advℜqm×1BiGISIS.

So, the difference of G2 from G1 is given by Eq. (7). (7) |PrSG1−PrSG2|≤qH022dH0+qH122dH1+qsend+qexecute22dχ+2Advℜqm×1BiGISIS

G3:Unlike G2, this game models Ă’s manipulation of the m´’s mobile device with the Corrupt-m´(m´i) query. This attack assumes that Ă has taken over the device in various ways. Therefore, Ă can access sensitive information and use this information for authentication. In G3, Ă obtains pidm′,dm´ with the Corrupt-m´(m´i) query. Then, he/she tries to guess m´’s identity (ID) and password with the send query. With each send query, the possible user ID-password pair is eliminated. Let’s examine Ă’s success in G3.

Let the number of send queries be qsend and the dimension of the passwords and ID’s dictionaries be dpw and did, respectively. With the Corrupt-m´(m´i) query, the success probability of Ă in G3 is given by Eq. (8). (8) |PrSG2−PrSG3|≤qsenddiddpw

G4:Unlike G3, Ă gets the internal user role. With the use of this role, G4 is modeled to make an SLA attack for ś. Finally, Ă makes the send query to obtain the static secret key of ś. Let’s examine Ă’s success in G4.

In the BiGISIS.PAKE, the shared key is generated without sending additional signals, with the MSB reconciliation mechanism, defined in Definition 2, based on parameter limits. As expressed in the Fig. 2, no signal is sent for m´ or ś . Therefore, SLA cannot be made by using public keys xm´,pm′. The success probability of Ă in G4 is given by Eq. (9). (9) |PrSG3−PrSG4|=0

Note 1 Note that it was stated that if a BiGISIS problem-based key-sharing idea uses a reconciliation solution without sending additional information, SLA cannot be applicable to the constructed scheme. For detailed security proofs, we refer to Bindel, Stebila & Veitch (2021), Qin et al. (2022).

G5:Unlike G4, Ă gains an ability to make the Corrupt-ś (śj) query. With this query, Ă gets the static secret key of ś (sśT) from the secure database of the user in ś . If the success probability of Ă is negligible in G5, the concept of PFS is captured (Dabra, Bala & Kumari, 2020). In summary, Ă aims to generate the previously shared session keys using the parameters obtained by the Corrupt-ś (śj), thereby breaking PFS. Let’s examine Ă’s success in G5.

In this game, the main purpose of Ă is to break PFS by calculating the shared session key skm´ś = skś m´ given in Fig. 2. Ă can capture various parameters in the following ways.

• With the Corrupt-ś (śj) query: pidm´,PWm´,sśT∗,sśT

• With the excute query: pidm´,xm´∗, xśT,αś, {cidm´, αm´}

Using these parameters, Ă computes xm´=xm´∗−PWm´. Then, Ă needs sśT and rśT values to generate ψś and ψm´′. However, to compute skm´ś or skś m´, Ă should additionally get rśT and rm´. These components can only be obtained by solving the BiGISIS problem. As a result of Conclusion 1, Eq. (10) is procured. (10) |PrSG4−PrSG5|≤Advℜqm×1BiGISIS︷<negln

G6:Assume that Ă tried and failed every possible way to break the BiGISIS.PAKE scheme’s security in G0, …, G5. Ă does the test query and tries to guess procured b. In G6, the CDF-Zipf model is also considered to characterize the password distribution. So, the main difference of G6 from G5 is re-valuated with Eq. (11) by considering CDF-Zipf assumptions. (11) PrSG6≤PrCorrectpw︷C′⋅nopf+PrSG5|¬Correctpw︷1/2Pr¬Correctpw︷1−C′⋅nopf≤1/21+C′⋅nopf

By considering Eqs. (6)–(11), Eq. (4) is rewritten and Eq. (12) is obtained. (12) AdvĂBiGISIS.PAKE≤2|PrSG0−12|=2|PrSG0−PrSG6|=2|PrSG0−PrSG1=G2|︷≤qH022dH0+qH122dH1+qsend+qexecute22dχ+|PrSG2−PrSG3|︷≤qsenddiddpw+|PrSG3−PrSG4|︷0+|PrSG4−PrSG5|︷Advℜqm×1BiGISIS+|PrSG5−PrSG6|︷1/21+C′⋅nopf≤qH02dH0+qH12dH1+qsend+qexecute2dχ+2qsenddiddpw+4Advℜqm×1BiGISIS+C′⋅nopf<negl(n)

So, Theorem 1 is satisfied by Eq. (12).

Security proofs

In the game-based security analysis of the proposed PAKE, resistance analysis against; eavesdropping (G1), hash functions simulation (G2), MBA (G3), SLA (G4), and PFS (G5) attacks are discussed. Let’s examine some security-related concepts.

• Integrity: In the proposed PAKE, authentication checks are done with the help of the controls defined in steps 27 and 40. If these steps (αm´′=?αm´ where αm´=H0idm´,sm´,cidm´,xm´,ψm´,xśT,ψś′,αś and αś=?H0xm´,xśT,ψś′) are satisfied correctly, then ś and m´ will be valid. The components examined in authentication include all messages exchanged between the two parties and are generated with a secure hash function (H0). Thus, the protocol satisfies the integrity and authentication of the message.

• Impersonation Attack: In the proposed idea, the authentication checks are done by looking αś=?H0xm´,xśT,ψś′ and αm´′=H0idm´,sm´,cidm´,xm´,ψm´′,xśT,ψś,αś. When Ă tries to impersonate the parties, he/she needs to have idm´,sm´,ψś,ψm´′. Ă can not obtain idm´ with the anonymity property and get sm´,ψś,ψm´′ get since the hardness of BiGISIS. So, the impersonation attempts of Ă do not work against the proposed PAKE.

• MitM Attack: With MitM, Ă can obtain pidm´,xm´∗, xśT,αś, and {cidm´, αm´}. By using these components, Ă cannot impersonate parties due to the hash functions (H{i=0,1,2}), authentication checks, and no polynomial-time algorithm to solve BiGISIS problem.

• Offline PDA: Let Ă captures the mobile device at the registration stage and obtains dm´=dm´∗⊕H0idm´,t⊕H0idm´,pwm´ and vm´ = H0(idm´, pwm´, dm´) given in Fig. 1. In this case, even if Ă guesses pwm´ correctly, Ă cannot complete the registration phase without idm´. Overall, it is resistant to offline dictionary attacks.

• KKS: In the constructed PAKE, shared key is computed with skś m´=H0idm´,cidm´,xm´,ψm´′,αm´,xśT,ψś,αś. Each component varies depending on the session and unique parties. In addition, the reusable key feature is provided. So, the attacker cannot obtain information about previous or subsequent sessions and KKS is provided.

Comparison

The parameters are chosen based on Ding, Cheng & Qin (2022) and the reconciliation’s upper limit. The calculation model of Islam & Basu (2021) is used to obtain the running time comparison results of Ding, Cheng & Qin (2022) and the BiGISIS.PAKE schemes. In Ding, Cheng & Qin (2022), Sony Xperia XP with Qualcomm MSM8996 1.5 GHz CPU and 3 GB RAM was used for m´. For ś, the experiments are based on two 32-core computers with 3 GHz Intel Xeon E5-2620 CPUs and 64GB RAM. For n = 512, the approximate average run times of main computations are presented in Table 1 and are determined with 10,000 times running results.

Table 1 The BiGISIS.PAKE approximate average run times in milliseconds (ms).

	T H 0	T H 1	T H 2	T samp	T mul	T msb	
Server	0, 1344	1, 8422	2, 7113	1, 4778	7, 5872	1, 408	
User	0, 4392	10, 2406	11, 2723	8, 8761	11, 1852	6, 3641	
Notes.

THi=0,1,2, Tmsb: Running time for hash and reconciliation functions.

Tsamp: Running time for sampling from χ.

Tmul: Running time for multiplication in ℜqm×1.

To determine approximate average times of m´ and ś , the main components of each phase are defined in Table 2.

Table 2 Comparison components.

		Registration	Login& Password-Based mutual authentication	Password update	
Ding, Cheng & Qin (2022)	m´	3H0, 1H1	6H0, 1H1, 2H2, 3χ, 4x, 1Sig, 2Mod2	4H0, 2H1	
ś	2H0, 1χ	5H0, 2H2, 2χ, 4x, 1Sig, 2Mod2	-	
BiGISIS.PAKE	m´	3H0, 1H1	6H0, 1H1, 2H2, 6χ, 6x, 2MSB	4H0, 2H1	
ś	2H0, 1χ	5H0, 2H2, 4χ, 5x, 2MSB	-	
Notes.

For example, BiGISIS.PAKE needs 3 H0 and 1 H1 hash function for m´ during in registration.

χ: Represents the parameters selected from the discrete gaussian distribution.

Sig, Mod2, and MSB represent the number of reconciliation functions.

x represents the number of multiplications needed in the step.

By using Table 1 and Table 2, the running times of the proposed scheme are presented in Table 3 by giving phase-wise results.

To make a comparison, the running times of Ding, Cheng & Qin (2022) phases are also determined by using the same calculation method. Note that in the current literature, two four-phase PAKE have been proposed with similar design ideas. Since Dabra, Bala & Kumari (2020) was proven to be vulnerable to SLA in Ding, Cheng & Qin (2022), we only performed the comparison with Ding, Cheng & Qin (2022). Let’s explain how the calculations in Table 3 were made for the BiGISIS scheme.

Table 3 Approximate average run times of The BiGISIS.PAKE phases in ms.

		Ruoyu.PAKE⋆ (Ding, Cheng & Qin, 2022)	BiGISIS.PAKE⋆∗	
Registration	m´	4, 5178	11, 5582	
	ś	0, 7309	1, 7466	
Password-Based Mutual Authentication	m´	41, 5894	168, 5164	
	ś	14, 1124	52, 7578	
Password Update	m´	8, 1572	22, 238	
	ś	−	−	
Total	m´	54, 2644	202, 3126	
	ś	14, 8433	54, 5044	
Notes.

⋆∗: {n = 512, σ = 3, 192, β = 72, 2267, q = 1073479709}.

∗: {mlogm=3, 2}

−: No component.

Figure 4 Lattice-based PAKE schemes in terms of similar features.

• Registration: For m´, 3 × TH0 + 1 × TH1 = 3 × 0, 4392 + 10, 2406 ≈ 11, 5582 is calculated. For ś , 2 × TH0 + 1 × Tsamp(χ) = 2 × 0, 1344 + 1, 4778 ≈ 1, 7466 is obtained.

• Login&Password-Based Mutual Authentication: For m´, 6 × TH0 + 1 × TH1 + 2 × TH2 + 6 × Tsamp(χ) + 6 × Tmul(×) + 2 × Tmsb(MSB) = 6 × 0, 4392 + 1 × 10, 2406 + 2 × 11, 2723 + 6 × 8, 8761 + 6 × 11, 1852 + 2 × 6, 3641 ≈ 168, 5164is computed. For ś , 5 × TH0 + 2 × TH2 + 4 × Tsamp(χ) + 5 × Tmul(×) + 2 × Tmsb(MSB) = 5 × 0, 1344 + 2 × 2, 7113 + 4 × 1, 4778 + 5 × 7, 5872 + 2 × 1, 408 ≈ 52, 7578 is determined.

• Password Update: 4 × TH0 + 2 × TH1 = 4 × 0, 4392 + 2 × 10, 2406 ≈ 22, 238 is calculated for m´.

Note 2 Unlike (Ding, Cheng & Qin, 2022), TSig components are not included in the calculation because the proposed PAKE does not use additional information.

Table 4 Comparison of four-phase lattice-based PAKE protocols.

	Hard problem	Number of party	Number of arithmetic operation −	Number of flow*	Number of hash	Reconciliation bound	Reconciliation components	Reusable key	Used method for SLA resistance	Anonymity	Security model	PFS	
Dabra, Bala & Kumari (2020)	RLWE	2	3 ×
3 +	9	3	q>164β2n23+2βn12	Cha(), Mod2()	×	Direct Public Key Validation From Zero-Knowledge Authentication	✓	RoR	✓	
Ding, Cheng & Qin (2022)	RLWE	2	2 ×
2 +	9	3	q>16α3n52+αn12	Cha(), Mod2()	✓	Practical Randomized KE+	✓	RoR	✓	
Islam & Basu (2021)	LWE	3	1×	10	3	q>16β2n23	Cha(), Mod2()	×	×	×	ROM	✓	
Li, Wang & Morais (2020)	LWE	2	− −	2	4	For n,m≥nlogq, inner product <e, k > should be negligible.	×	×	×	×	ROM	×	
BiGISIS.PAKE	BiGISIS	2	2 ×
5 +/-	9	3	∥kś − km´ ∥  ≤ 2(8ϑ + β)	MSB()	✓	BiP	✓	RoR	✓	
Notes.

*Total number of passes for all stages, ×, Multiplication; +, Addition; −, Subtraction; − −,No computation; −, Number of arithmetic operations used in key component calculation; ROM, Random Oracle Model) +: According to Bindel, Stebila & Veitch (2021) and Qin et al. (2022), because of the used reconciliation, SLA is possible.

In Table 3, the BiP components are the main reason for high running times for the BiGISIS.PAKE. The other fundamental factors that cause differences are visualized in Fig. 4 and summarized in Table 4.

With Fig. 4, the main differences of similar lattice-based PAKE schemes are detailed with a component-base comparison in terms of reconciliation method, hard problem, key generation idea, and SLA resistance solutions. This analysis answers the question of how similar literature PAKE schemes are differentiated from each other.

According to Table 4, the proposed BiGISIS.PAKE is different from the other PAKEs due to the hard problem, reconciliation structure, and the method for SLA resistance. Even if the BiGISIS.PAKE has much more arithmetic operations and running time due to the BiP method used for SLA resistance and MSB reconciliation function; it does not require any signal information to obtain the shared key. In Dabra, Bala & Kumari (2020), Ding, Cheng & Qin, (2022) and Kumar et al. (2023) the Ding reconciliation method, defined with Cha() and Mod2() functions, was added as a solution of reconciliation problem. In this approach, the idea of sending additional information about the key component was used to generate the shared key. In Bindel, Stebila & Veitch (2021), Qin et al. (2022), it was stated that schemes that utilize additional information-based reconciliation will be vulnerable against SLA. Therefore, due to the selected structures, BiGISIS.PAKE is the first lattice-based PAKE resistant to SLA attacks with the anonymity and reusable key in the literature.

Conclusion

In this article, a novel lattice-based PAKE scheme, BiGISIS.PAKE, based on the hardness assumption of the BiGISIS problem, is proposed. The anonymity is guaranteed by the one-way property of the hash functions and the inability to obtain static secret keys during the login&password-based mutual authentication phase. BiP approach is added to get the reusable key feature to reduce key generation time. The security analysis of the four-phase BiGISIS.PAKE is performed in the RoR model to show the resistance against eavesdropping, SLA, hash function simulation, and MBA. With the help of additional queries, it is shown that it provides PFS and integrity features. The robustness analysis of the proposed PAKE against impersonation, MitM, KKS, and offline PDA is also presented. Even if the required running time is high due to the reusable key, the public key, generated once, can be used more than once. To the best of our knowledge, the proposed PAKE is the first lattice-based PAKE that satisfies anonymity and reusable key with SLA resistance.

Additional Information and Declarations

Competing Interests

Author Contributions

Data Availability

Sedat Akleylek is a Section Editor of PeerJ Computer Science.

Kübra Seyhan conceived and designed the experiments, performed the experiments, analyzed the data, performed the computation work, prepared figures and/or tables, and approved the final draft.

Sedat Akleylek conceived and designed the experiments, performed the experiments, analyzed the data, performed the computation work, authored or reviewed drafts of the article, and approved the final draft.

The following information was supplied regarding data availability:

This article did not use raw data or code.

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
