# Peer review of "A new lattice-based password authenticated key exchange scheme with anonymity and reusable key"

_PeerJ Computer Science, doi:10.7717/peerj-cs.1791_

## Round 0.1 · original submission · Major Revisions

The contributions need to be clarified. Consideration of all the comments is a must. The consideration could be by modification or just replying to a comment with justified answer. Usage of the recommended references by the reviewers is not mandatory; they are just suggestions as examples.

Reviewer 1 ·

Basic reporting

This paper attempts to provide a new lattice-based password authenticated key exchange (PAKE) scheme, based on a BiGISIS problem, aiming to offer a solution for post-quantum secure PAKE scheme. Here is a critical review based on the given information:
• The BiGISIS problem, which this paper heavily relies upon, was only defined in 2018, and one may argue whether this problem has undergone enough scrutiny and validation in the cryptographic community to base new schemes on it securely. If it hasn’t been sufficiently validated by the cryptographic community or lacks a sound theoretical basis, any resultant PAKE schemes may also be inherently flawed or insecure.
• The comparative analysis seems to underscore the uniqueness of this PAKE scheme over existing lattice-based schemes but does not sufficiently detail how this scheme outperforms existing ones in terms of security, efficiency, and applicability. It’s crucial to critically assess whether the novel features introduced truly represent advancements in the field or merely introduce unnecessary complexity and potential vulnerabilities.

Experimental design

• The four-phase structure of the BiGISIS.PAKE seems rather intricate, involving setup, registration, login&password-based mutual authentication, and password update phases. It is critical to question whether such complexity is warranted and if it might lead to implementation errors and potential security vulnerabilities, particularly in real-world, large-scale applications.
• While the paper claims to provide anonymity and reusable key features, it doesn’t offer extensive details or robust proof regarding how these features are securely implemented. It is essential to critically assess whether the implementation of these features might inadvertently introduce new vulnerabilities or compromise the security of the scheme.• The paper claims security by relying on the RoR model, but the absence of reveal queries and reliance on standard queries in the model might not capture all potential security issues and attack vectors. It’s critical to ask whether the security analysis is comprehensive enough and whether it considers all relevant threat models and potential attack vectors, particularly in post-quantum scenarios.
• Despite its claims, the paper concedes that the proposed PAKE has more runtime compared to similar lattice-based PAKE schemes due to its reusable key feature. It is stated that this is negligible since the shared session key can be used more than once, but this raises critical questions about the practical efficiency and performance of this scheme, especially in scenarios where rapid key generation is essential.
• The paper posits this scheme is suitable for mobile device-server communication security due to the BiGISIS.PAKE’s four-phase structure and additional features. However, it fails to provide empirical data, real-world testing results, or in-depth analysis to substantiate this claim. The practical feasibility and performance of the proposed scheme in real-world applications, particularly in diverse and possibly constrained environments like mobile devices, remain uncertain.

Validity of the findings

• While the paper concludes with several assertions about the proposed scheme's security features and advantages, it is crucial to examine whether the provided security analysis and discussions sufficiently validate these conclusions. It is imperative that all assertions made are strongly supported by rigorous theoretical proofs, empirical data, or comprehensive analysis to ensure the reliability and soundness of the proposed scheme.

Additional comments

• The paper seems to lack conciseness and clarity in representing the fundamental concepts, results, and implications of the proposed scheme. A more structured and streamlined presentation of the key ideas, implementation details, and security proofs would be critical for the readership to gain a clear understanding of the contributions and implications of this work.

Reviewer 2 ·

Basic reporting

this area is OK

Experimental design

experimental design is OK

Validity of the findings

cannot really tell, i cannot replicate the tests, but they seem OK

Additional comments

Fascinating and timely article. It deserves publication, and I am recommending acceptance with corrections. Some issues require your attention. I list these corrections below as feedback/comments, and I look forward to reading this article's updated version.

-- Have you considered the effect of new (future) quantum computers on the findings of your study? See: ‘Red Teaming Generative AI/NLP, the BB84 quantum cryptography protocol and the NIST-approved Quantum-Resistant Cryptographic Algorithms’: https://arxiv.org/abs/2310.04425. It would be engaging to read your thoughts on this emerging area of risk, and given that its coming in the future, your article would stand strong with time if you include a short discussion on this topic. I would also be very interested to see a few sentences reviewing and comparing your work in relation to this very recent study in a related topic.

-- You have figures in the conclusion, this is not a usual practice. You should check if all the things discussed in the introduction are also discussed in the conclusion. because the introduction is much longer than the conclusion. Usually, these sections are comparable in length. If you think you have covered everything, that’s OK, but to mention that conclusion is the best chapter to outline your key findings and key conclusions. So, it would be best if you used this chapter to make your article more readable, and since most readers would focus a great deal of their attention on the conclusion, this section should make the key conclusions more visible (and hence more interesting).

Reviewer 3 ·

Basic reporting

In this article, the authors propose a novel password-authenticated key exchange (PAKE) scheme based on bilateral generalization inhomogeneous short integer solution (BiGISIS) for post-quantum era security. The authors employ the bilateral-pasteurization (BiP) approach to derive the reusable shared key, and anonymity is achieved through the utilization of identity components in the hash functions. The reusable key structure reduces the time required for key generation, while anonymity serves as a deterrent against unauthorized user login attempts. The security analysis follows the real-or-random (RoR) model assumptions. To enhance the quality of the manuscript, it is recommended that the authors address the following issues.

Strengths:
* * *
1. The formulas and symbols have been properly explained.
2. The research on anti-quantum attacks is promising.

Weaknesses:
* * *
1. The organization and analysis of the comparison is not clear.
2. The summary of related work is vague.

Comments:
* * *
1. The summary of related work lack analysis and relevance to the research topic of this article. It is strongly recommended that the authors reorganize and analyze the related work to align it with the research focus.

2. The contributions listed in Section 1.2 are too trivial. I suggest the authors to summarize and consolidate them into 3-4 substantial work summaries.

3. Some citations are too brief, such as “It is also proved in (Bindel et al, 2021; Qin et al, 2022) that” in page 3 line 110. It is difficult for readers to immediately understand the authors’ intended meaning. Suggest the authors to paraphrase the content of essential citations.

4. In section 3, the figures 1,2,3 and the descriptions of the steps are indistinguishable to the readers. Therefore, the readers cannot learn the analysis of steps, parameters, computations, and other details from the descriptions. I strongly recommend the authors to analyze and explain the steps in the figures within the descriptions to enhance readability.

5. An issue is related to the implicit assumption made about the threat model. Suggest the authors to explicitly explain what capabilities are allowed to the attacker against PAKE schemes. They mainly mentioned some attacks, but only these attacks are not complete. It is strongly recommended that the authors explicitly define a security model and may refer to the following models. A separate subsection shall be used to discuss the threat model.

“A Survey of Important Issues in Quantum Computing and Communications”. IEEE Communications Surveys & Tutorials, 2023.
"Quantum2FA: Efficient Quantum-Resistant Two-Factor Authentication Scheme for Mobile Devices", IEEE Transactions on Dependable and Secure Computing, 2022
"Quantum-Safe Round-Optimal Password Authentication for Mobile Devices", IEEE Transactions on Dependable and Secure Computing, 2020

6. The security formulation of password distribution is not accurate. Recent research has shown that user-chosen passwords (which are termed as “weak secrets” in PAKE protocols) follow the Zipf's law, a vastly different distribution from the uniform distribution. Actually, the size D of password space is generally much constrained: users will not use the whole space of passwords, but a rather small space of the allowed character space, as shown in the following research [IEEE TIFS17].

"QPause: Quantum-resistant password-protected data outsourcing for cloud storage", IEEE transactions on Services Computing (TSC), 2023, Doi: 10.1109/TSC.2023.3331000
"Zipf's Law in Passwords," IEEE Transactions on Information Forensics and Security, 2017.
"Targeted online password guessing: An underestimated threat,” in Proc. ACM CCS 2016, pp. 1242–1254.

As for how to use the above mentioned Zipf-based security formulation, please see:

"Provably Secure Fine-Grained Data Access Control over Multiple Cloud Servers in Mobile Cloud Computing Based Healthcare Applications", IEEE Transactions on Industrial Informatics, 2018.
"Lightweight and Physically Secure Anonymous Mutual Authentication Protocol for Real-Time Data Access in IWSN", IEEE Transactions on Industrial Informatics, 2018.


7. In Section 5, it is recommended that the authors reformat the tables 1,2 to enhance readability for comparative analysis. Additionally, is it fair to only compare a single related work (Ding et al, 2022)? Are there any other works that can be analyzed?

8. It is recommended that the authors analyze the reasons for efficiency improvement and provide insights into the related works while comparing the costs.

9. Figure 4 is unclear, and please check if the font is in compliance with the specifications. The font size in tables 1, 2, 3, and 4, as well as in figures 2 and 3, is too small and affects readability.

In all, I suggest a "Major Revision''.

Experimental design

see above

Validity of the findings

see above

Additional comments

see above

·

Basic reporting

Clear and unambiguous, professional English used throughout.

Experimental design

Research question well defined, relevant & meaningful. It is stated how research fills an identified knowledge gap.

Validity of the findings

Conclusions are well stated, linked to original research question & limited to supporting results.

Additional comments

Authors should revise manuscript according to comments below.
1. Figures are not clear enough, and there are no enough explaination or description about figures. Please adjust resolution of figures and add description about figures in detail.
2. Authors should define "reusable key" and "anonymity" in manuscript including related works.

---

## Round 0.2 · accepted · Accept

Congrats on your paper acceptance!

Reviewer 1 ·

Basic reporting

The authors have revised well. All my suggestions and comments were addressed appropriately. I have no further comments.

Experimental design

The authors have revised well. All my suggestions and comments were addressed appropriately. I have no further comments.

Validity of the findings

The authors have revised well. All my suggestions and comments were addressed appropriately. I have no further comments.

Additional comments

The authors have revised well. All my suggestions and comments were addressed appropriately. I have no further comments.

Reviewer 2 ·

Basic reporting

Corrections are OK.

Experimental design

Corrections are OK.

Validity of the findings

Corrections are OK.

Additional comments

Corrections are OK.

Reviewer 3 ·

Basic reporting

The revised version has addressed most of my concerns, and I suggest an acceptance.

Experimental design

The revised version has addressed most of my concerns, and I suggest an acceptance.

Validity of the findings

The revised version has addressed most of my concerns, and I suggest an acceptance.

Additional comments

The revised version has addressed most of my concerns, and I suggest an acceptance.